# Shedding New Light on Mountainous Forest Growth: A Cross-Scale Evaluation of the Effects of Topographic Illumination Correction on 25 Years of Forest Cover Change across Nepal

Jamon Van Den Hoek [1,*], Alexander C. Smith [1], Kaspar Hurni [2,3], Sumeet Saksena [3] and Jefferson Fox [3]

1 Geography Program, College of Earth, Ocean, and Atmospheric Sciences, Oregon State University, Corvallis, OR 97331, USA; smitale3@oregonstate.edu

2 Centre for Development and Environment, University of Bern, Mittelstrasse 43, CH-3012 Bern, Switzerland; kaspar.hurni@unibe.ch

3 East-West Center, 1601 East-West Road, Honolulu, HI 96848, USA; saksenaS@eastwestcenter.org (S.S.); foxJ@eastwestcenter.org (J.F.)

* Correspondence: jamon.vandenhoek@oregonstate.edu

**Abstract:** Accurate remote sensing of mountainous forest cover change is important for myriad social and ecological reasons, but is challenged by topographic and illumination conditions that can affect detection of forests. Several topographic illumination correction (TIC) approaches have been developed to mitigate these effects, but existing research has focused mostly on whether TIC improves forest cover classification accuracy and has usually found only marginal gains. However, the beneficial effects of TIC may go well beyond accuracy since TIC promises to improve detection of low illuminated forest cover and thereby normalize measurements of the amount, geographic distribution, and rate of forest cover change regardless of illumination. To assess the effects of TIC on the extent and geographic distribution of forest cover change, in addition to classification accuracy, we mapped forest cover across mountainous Nepal using a 25-year (1992–2016) gap-filled Landsat time series in two ways—with and without TIC (i.e., nonTIC)—and classified annual forest cover using a Random Forest classifier. We found that TIC modestly increased classifier accuracy and produced more conservative estimates of net forest cover change across Nepal (−5.2% from 1992–2016). TIC also resulted in a more even distribution of forest cover gain across Nepal with 3–5% more net gain and 4–6% more regenerated forest in the least illuminated regions. These results show that TIC helped to normalize forest cover change across varying illumination conditions with particular benefits for detecting mountainous forest cover gain. We encourage the use of TIC for satellite remote sensing detection of long-term mountainous forest cover change.

**Keywords:** topographic correction; Landsat; time series; LandTrendr; forest growth; land cover change; Google Earth Engine

## 1. Introduction

Changes in global temperature and precipitation patterns are affecting the world's mountains at an unprecedented rate [1–4]. These changes have unfolding consequences for the estimated 23–28% of global forests in mountainous areas as well as the over 700 million people who depend on mountainous forests for timber and non-timber forest products for their livelihoods, food security, and sustainable development [5–7]. Mountainous forests also provide critical habitat for endemic and endangered species, often have an essential role in national economic welfare [8,9], and offer essential ecosystem services such as carbon sequestration, air purification, soil nutrient cycling, and slope stabilization [10]. Increasingly, processes of deforestation and forest regeneration are monitored using a long-term and continuous (annual or sub-annual) perspective [11–14], which means that additional rigor must

be given to measuring and mitigating intra- and inter-annual variation in image acquisition conditions [15,16], especially in mountainous environments [17,18]. For example, frequent cloud cover and rugged terrain lead to variability in surface reflectance that is unrelated to land cover phenology, trends, or change in land cover type [19–21], and illumination conditions that vary with solar geometry (i.e., solar azimuth and zenith) and terrain orientation (i.e., slope and aspect) affect exitant radiance in spatially and temporally variable ways [18].

The goal of topographic illumination correction (TIC) is to reduce the variation in apparent feature reflectance to better isolate legitimate differences in reflectance [17,22,23], and several TIC approaches having been developed in recent years [18,22,24,25]. Most forest remote sensing studies to date (Table 1) have focused on evaluating the effect of TIC on classification accuracy using single-date satellite imagery (e.g., [17,26,27]) and have generally found that TIC improved overall forest cover classification accuracy by 1–3%. Studies on remote sensing of multi-date forest cover change have found similarly slight improvements in change detection accuracy with TIC [20,28–34].

**Table 1.** Review of recent research on topographic illumination correction effect on Landsat-based measures of land cover, forest cover, and forest cover change. PBM: Pixel Based Minnaert, PBC: Pixel-Based C-correction, SCS: Sun-Canopy-Sensor, S-E: Statistical Empirical, VECA: Variable Empirical Coefficient Algorithm.

| Image Frequency | Theme | Correction Approach(es) | Study Area | Citation |
|---|---|---|---|---|
| Single date | Forest Cover | Band Ratioing, Cosine Correction, PBM, and PBC | Carpathian Mountains, Romania | [35] |
| | Vegetation Cover | Illumination Condition Weighted Mean, Minnaert, C-correction | Cabañeros National Park, Spain | [27] |
| | | Band Ratioing, Cosine Correction, Pixel Based Minnaert (PBM), and Pixel-Based C-correction (PBC) | Carpathian Mountains, Romania | [36] |
| | Land Cover | Teillet-Regression, Cosine Correction, Sun-Canopy-Sensor (SCS), SCS+C, Minnaert, Minnaert-SCS | Shanxi Province, China | [26] |
| Multi-date | Forest Cover Change | C-corrected and modified C-correction | Peloponnese Peninsula, Greece | [33] |
| | | C-correction, Statistical Empirical (S-E) and Variable Empirical Coefficient Algorithm (VECA) | Central Adirondack Mountains, United States | [28] |
| | | Bin Tan | Tennessee, California, Utah, and Colorado, United States | [31] |
| | | PBM | Carpathian Ecoregion, Romania | [32] |
| | | C-correction, Improved Cosine, Minnaert, S–E, and VECA | Dong Phayayen-Khao Yai Forest Complex, Thailand | [29] |
| | | Bin Tan, C-correction, Minnaert with slope, S-E, SCS, and VECA | Nepal | [37] |
| Time series | Vegetation Cover Change | Lambertian and C-correction | Ebro Valley, Spain | [34] |
| | Forest Cover Change | Semi-empirical C-correction | Taita Hills, Kenya | [38] |
| | | SCS | Southwest British Columbia, Canada | [39] |
| | | C-correction | Bago Mountains, Myanmar | [30] |

Despite recent scholarship, several underexplored aspects of TIC's effects on forest cover and forest cover change mapping remain. While the marginal effect of TIC on forest cover classification accuracy is well-established, there has been very little attention towards how TIC affects the actual areal measurements of forest cover and change over long periods of time. TIC should reduce the apparent bias towards detecting forest cover that was better illuminated at the time of satellite image acquisition and, in turn, normalize measurements of the extent and rate of forest cover change across the landscape regardless of illumination. Given a hypothetically uniform distribution of forest cover gain across a landscape, more forest cover gain would be expected to be detected in less illuminated regions with TIC

than without. Further, since the geographic distribution and rate of forest cover change may vary over time, TIC may thus result in an over- or underestimate of forest cover or forest cover gain compared to uncorrected data depending on the location and the stage of forest regeneration. However, the effect of TIC on normalizing forest cover change has never been directly addressed. Moreover, the routine examination of TIC effects at validation sites that are all contained within a relatively small study area—often a single Landsat footprint—impedes understanding of how TIC effects on forest cover change measurements may vary across a landscape's diverse topographic conditions.

In moving beyond an examination of the effects of TIC on classification accuracy, this is the first study to explicitly examine the effects of TIC on measures of forest cover and change and describe how those effects vary over time and space. The specific goal of this study is to assess the effects of TIC on the extent, geographic distribution, and type of forest cover change. To achieve this goal, we created a nationwide, 25-year (1992–2016) gap-filled Landsat time-series dataset across the topographically complex landscape of Nepal in two ways—with TIC and without TIC (i.e., nonTIC). We mapped annual forest cover across the country using a Random Forest classifier with TIC and nonTIC approaches and summarized results at the national level as well as by physiographic zone and illumination condition. Our specific objectives are:

1.  Quantify differences in forest cover classification accuracy using TIC and non-topographic illumination corrected (nonTIC) data;
2.  Quantify differences in the extent and geographic distribution of forest cover change with TIC and nonTIC approaches; and
3.  Quantify differences in type of forest cover change (e.g., regenerated or lost forest cover) using TIC and nonTIC approaches.

The results of this study offer a novel and comprehensive assessment of the effects of TIC on measures of the extent, geographic distribution, and type of forest cover change with practical relevance for forest inventory mapping, forest management, and forest carbon monitoring in mountainous landscapes around the world.

## 2. Materials and Methods

### 2.1. Study Area

Extending over 147,181 km$^2$, Nepal is predominantly a mountainous country composed of three physiographic zones as defined by the Department of Survey Nepal in 1988 (http://rds.icimod.org/Home/DataDetail?metadataId=1597, accessed on 14 November 2020) (Figure 1): the Mountains (>3000 masl, 35% of Nepal's total land area, 7% of population), Middle Hills (700–3000 masl, 35% of land area, 37% of population), and Terai (lowland plains < 700 masl, 30% of land area, 56% of population). Forest and non-forest are recorded across all three physiographic zones (Uddin et al., 2015). Nepal has an estimated 443 different tree species across forest types ranging from subalpine conifer and temperate broadleaf forests to tropical and subtropical conifer forests and subtropical broadleaved forests [40]. Over the past 3 decades, an extensive body of literature on satellite remote sensing-derived forest cover change in Nepal has been developed. The authors in [40–43] used remote sensing to measure forest cover at the national scale, and [44–48] measured forest change at village and watershed scales. While there is a lack of direct comparability between these studies, they point to a prominent trend of increasing forest cover across Nepal starting in the late 1980s and continuing until the most recent studies. Much of the forest cover expansion in the late 1980s and 1990s has been attributed to the implementation of community forest management, changes in rules on open grazing, expanded forest plantations, and reduced need for cooking fuel [49,50]. By the 2000s and 2010s, the effects of Nepal's 10-year-long civil war, specifically conflict-induced out-migration, as well as post-war economic diversification have been identified as drivers of forest cover gain [46,51–53]. Being subject to subtle changes in mountainous forest cover—widespread and prolonged forest regeneration amidst continued selected extraction of forest products—

Nepal is thus an ideal case study to assess the effect of TIC on remote sensing estimates of long-term forest cover change.

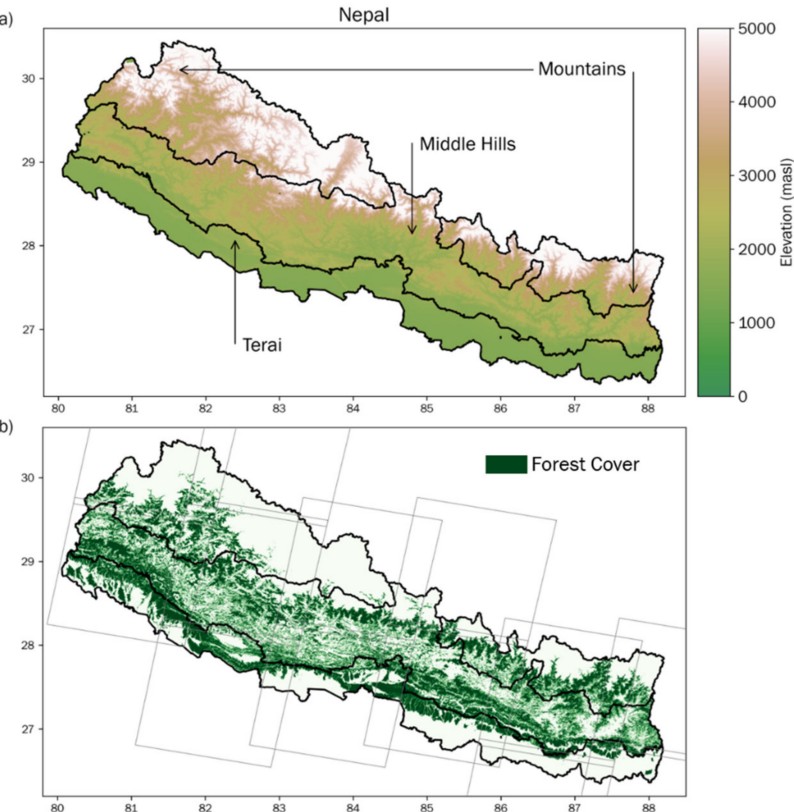

**Figure 1.** Nepal's (**a**) topography based on Shuttle Radar Topography Mission 30-m resolution Digital Elevation Model (Farr et al., 2007) with labeled physiographic zones (black), and (**b**) forest cover (shown in green) as measured by ICIMOD in 2010 (Uddin et al., 2015) overlaid by 13 Landsat WRS-2 path-row images used in the study (gray). Note that the inclusion of potentially snow- or ice-covered regions in the Middle Hills and Mountains zones have a negligible effect on forest cover mapping since snow and ice are removed with image pre-processing, as described below.

### 2.2. Data and Materials

We gathered 3305 USGS Landsat 5, 7, and 8 Tier 1 Surface Reflectance Climate Data Records (CDR) images at 13 WRS-2 footprints across Nepal (Figure 1b), collected during the peak greenness period of July through October from 1992 to 2016; 1992 was the first year with consistent Landsat coverage across Nepal, and 2016 was the last year with image availability when the study began. We also used corresponding Tier 1 Top of Atmosphere (TOA) Reflectance images solely for cloud masking purposes. We removed 299 Landsat SR images with more than 80% cloud cover based on CFmask [54] or with more than 80% cloud cover based on the Simple Cloud Score algorithm run on Landsat TOA images [55]), and removed an additional 1113 images because of an insufficient number of cloud-free forest pixels necessary for reliable estimation of TIC parameters or due to excessive atmospheric haze undetected by CFmask. This left 1893 images for analysis (Appendix A, Table A1). All image processing and analysis was carried out in Google Earth Engine [56].

To characterize topographic illumination (IL) conditions, we used the Shuttle Radar Topography Mission (SRTM) 30-m Digital Elevation Model (DEM) [57]. We measured the mean IL across all SR images in our time series. IL ranges from zero to one and is equivalent to the cosine of the solar incidence angle, which is based on a location's (i.e., pixel's) slope and aspect and the sun's position (solar zenith and azimuth) at the time of image acquisition (Equation (1); [18]). To assess how TIC may affect forest cover and change measurements differently due to IL conditions, we stratified Nepal's IL range into

deciles where IL stratum 1 includes the lowest (i.e., darkest) IL (0–0.09) and stratum 10 includes the highest (i.e., brightest) IL (0.90–1.00). IL tends to increase as elevation and slope decrease, and the majority of Nepal's land mass is in the brightest IL strata (7–10) (Figure 2). Considering the distribution by physiographic zone, the Mountains contain more (10.3%) of the least-illuminated strata (i.e., IL 1–4) than the Middle Hills (5.8%) and Terai (1.5%) combined (Figure 2e). Approximately 95% of the Terai is in IL stratum 8 or 9, showing that the Terai has a very uniform and high topographic illumination condition, while the Middle Hills and Mountains have much more diverse IL conditions.

$$IL = cos(i) = cos(Z) \times cos(s) + sin(Z) \times sin(s) \times cos(a - a') \tag{1}$$

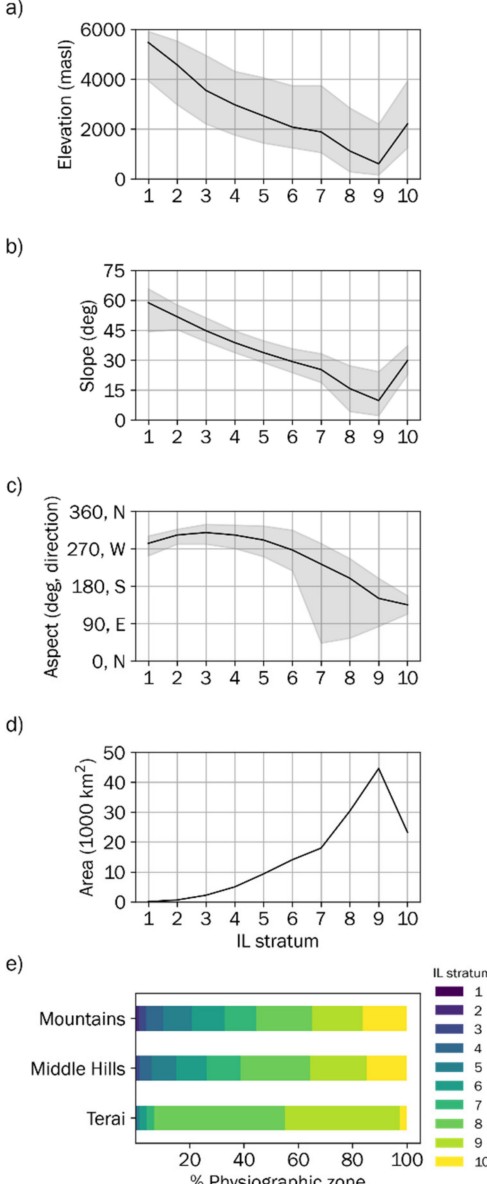

**Figure 2.** (**a**) Elevation, (**b**) slope, (**c**) aspect (black lines: median; grey ribbon: interquartile range), (**d**) area of each IL stratum, and (**e**) distribution of IL strata within Nepal's Mountain, Middle Hills, and Terai physiographic zones.

Calculation of IL. i = incident angle with respect to surface normal; Z = solar zenith angle; s = topographic slope; a = solar azimuth; a' = topographic aspect (azimuth).

### 2.3. Landsat Image Topographic Correction, Annual Compositing, and Trend Construction

We removed all clouds, cloud shadows, and snow cover identified using the CFmask algorithm (SR data) and the Simple Cloud Score algorithm (TOA data) [54]. We produced two separate Landsat time series for further analysis: one without topographic illumination correction—referred to as the nonTIC dataset henceforth—and another with topographic illumination correction—referred to as the TIC dataset henceforth. We smoothed the SRTM DEM with a 3 × 3 Gaussian low pass filter to mitigate the effect of surrounding topography and improve the performance of the TIC in areas with steep slopes [27]. Since our study's start in 1992 precludes use of ancillary data (e.g., MODIS, 2000–present) necessary to implement a physical or semi-physical TIC approach (e.g., [25,28]), we followed the recommendations by [37], who rigorously identified the best performing semi-empirical topographic illumination correction approach for each individual band by evaluating six methods at forested sites across four Landsat footprints within Nepal: C-correction [18], Sun-Canopy-Sensor and C-correction [58], Bin Tan [31], Statistical-Empirical (S–E; [18], Variable Empirical Coefficient Algorithm (VECA; [26]), and Minnaert with slope [59]. We ranked each correction's effect on each band using five criteria that consider how TIC affects the relationship between IL and reflectance (i.e., the coefficient of determination, and comparison of sunlit and shaded slopes) and the overall and variability of reflectance (i.e., interquartile range reduction, coefficient of variation, and relative difference in median reflectance); see [37] for full details on these band selection criteria. Following this approach, we found that VECA ranked best for the blue band and S–E ranked best for green, red, near infrared (NIR), and shortwave infrared (SWIR 1, SWIR 2) bands. We therefore applied an optimal band-wise TIC by applying VECA to the blue band and S–E to the remaining bands for each image in the TIC time series from 1992–2016.

For the nonTIC and TIC time series, we harmonized Landsat 8 spectral values with Landsat 5 and 7 using cross-sensor harmonization parameters provided by [60] and made annual composites using July–October growing season images. We determined the best pixel for each seasonal composite using three criteria previously used by [37]:

1.  Distance to the peak greenness date (1 September): Since pixels acquired closer to the peak greenness date are more helpful for discriminating forest cover, we assigned a weight of 1 to pixels acquired on September 1 and a value of 0.1 to pixels acquired at the beginning (1 July) or end (31 October) of the growing season following a Gaussian curve.

2.  Proximity of the pixel to clouds or cloud shadows: While CFmask is broadly effective at removing clouds and cloud shadows, some cloud or shadow pixels may remain, which would degrade the classification. We therefore weighted pixels by their Euclidean distance to clouds or cloud shadows using a Sigmoid function. Pixels more than 1500 m away from clouds or cloud shadows were given a weight of 1; for pixels closer than 1500 m, weights linearly decreased to 0 for pixels adjacent to clouds or cloud shadows.

3.  Quality of NIR reflectance: In a further effort to exclude shaded (low NIR value) or clouded pixels (high NIR value) in our classification, we calculated the median of all growing season images within a given year and assigned pixels with an NIR value equivalent to the median a weight of 1; pixels with the largest absolute deviation from the median were given a weight of 0. Using the median and the largest absolute deviation from the median, we linearly distributed weights between 0 and 1 to all other pixels based on their absolute deviation from the median.

We calculated the arithmetic mean of these three weights and selected the pixel with the highest mean weight for inclusion in a given year's growing season quality composite.

To estimate missing seasonal composite values due to clouds, shadows, and Landsat's SLC Error gaps, we fit pixel-level linear interpolations across both time series using the Google Earth implementation of LandTrendr [61]. LandTrendr identifies breakpoint dates associated with disturbance or regeneration, and linearly interpolates values between breakpoint dates [12,61]. We identified breakpoint dates using the SWIR 1 band and fit all

other Landsat spectral bands to these identified breakpoints. SWIR 1 was selected since it has been shown to be more sensitive to canopy moisture and forest structure compared to NIR or visible bands, for example [62,63].

### 2.4. Forest Cover Classifier Model Construction

Using linearly interpolated multispectral values (i.e., blue, green, red, NIR, SWIR 1, SWIR 2), we calculated Normalized Difference Vegetation Index (NDVI), Enhanced Vegetation Index (EVI), Normalized Burn Ratio (NBR), and Tasseled Cap Brightness, Greenness, Wetness, and Angle [64,65]. We also measured three topographic variables based on the SRTM DEM: elevation (masl), slope (degrees), and aspect (degrees). These 13 spectral and three topographic variables are commonly used in pixel-level forest cover mapping for Landsat time-series imagery [62,63,66–68]; may differently support forest detection with differences in condition, biomass, and illumination condition; effectively characterize differences between forest and non-forest sites in mountainous areas; and are broadly stable between TIC and nonTIC.

To generate an initial sample of forest and non-forest sites across Nepal, we conducted a stratified random sample of ICIMOD land cover data from 1990 and 2010 [41]. Since the purpose of this study was to assess the effects of topographic illumination correction on the accuracy, extent, and changes in classified forest cover maps, we did not preemptively remove any topographic regions, such as those above the tree line, from our sampling design. At each sample site, we visually interpreted the central and surrounding eight pixels using Landsat and very high-resolution reference imagery available in Google Earth. A site was considered to be forest if at least 50% of the central pixel was visually interpreted as being closed forest canopy, with higher interpretation confidence when surrounding pixels were similarly forested. A site was considered to be non-forest if it was visually interpreted to have less than 50% forest canopy closure. Using the 1990 ICIMOD data, we verified 45% of ICIMOD-classified forest (F) and 31.5% of non-forest (NF) samples with high confidence, and using the 2010 ICIMOD data, we verified 68% of ICIMOD-classified F and 36% of NF samples with high confidence. After verification, we had 2621 high-confidence samples composed of 935 F (35.7%) and 1686 NF (64.3%) samples (Figure 3a); this distribution mirrored the 2010 ICIMOD land cover distribution of 38.9% F and 61.1% NF. Samples were well distributed across the three physiographic zones with proportionately fewer F samples in the Mountains and IL strata 1 and 2 since much of these regions lie above the forest line (approximately 4000 masl (Figure 3b); zero forest sample sites were recorded above the forest line. Proportionate to IL stratum area, the number of F samples generally declined with increasing IL, whereas NF samples tended to increase (Figure 3c).

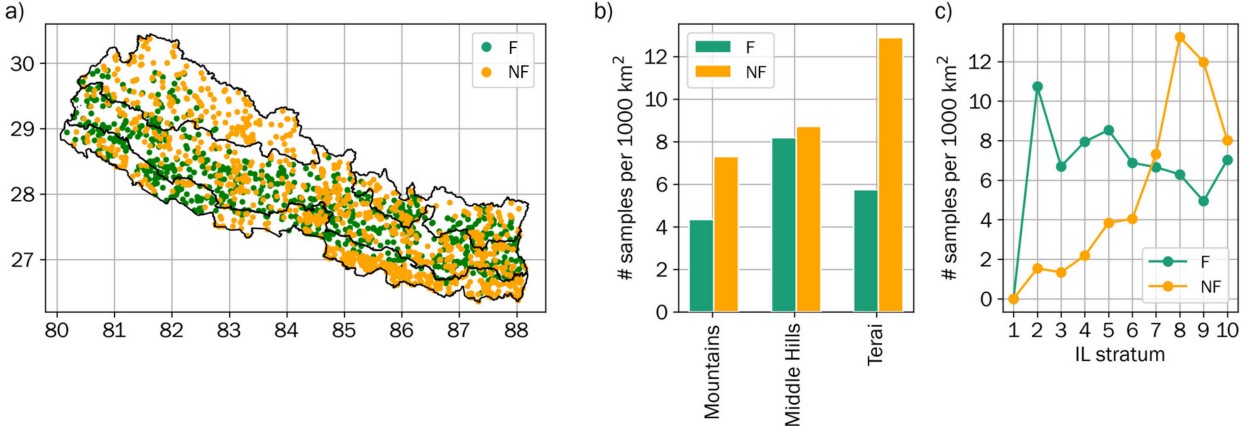

**Figure 3.** Distribution of forest (F, green) and non-forest (NF, orange) samples (**a**) across Nepal, (**b**) by dominant physiographic zones, and (**c**) by IL stratum.

Two distinct Random Forest classifier models were built in Google Earth Engine with one thousand decision trees in out-of-bag (OOB) mode, which supports a bootstrapped accuracy estimation approach that randomly subsamples the training and validation data without replacement. The models were trained using 16 spectral and topographic predictor variables (as described above) from TIC and nonTIC annual composites sampled at validated 1990 and 2010 sites labeled as "forest" (F) or "non-forest" (NF). The models differed only in that the nonTIC model used uncorrected spectral data while the TIC model used band-specific topographic illumination correction, as detailed above. The differences between nonTIC and TIC model accuracy are examined in Objective 1, differences in inter-annual and long-term forest cover change are examined in Objective 2, and differences in forest cover conversions (i.e., stability, regeneration, loss) are examined in Objective 3.

*Objective 1: Compare nonTIC and TIC Classification Accuracy*

We measured each model's accuracy using the OOB accuracy as well as validation, user's, and producer's accuracies using three-fold cross-validation of one-third samples used for testing and two-thirds reserved for validation. Our accuracy measures apply to forest cover maps for any given year in the study period. Since we harmonize our spectral data across all years and use linear interpolation to estimate values for locations with missing data on a given year, this helps ensure that our spectral measurements from one year are directly comparable to another year, and that our validation data collected in a given year support an accuracy assessment for the entire study period.

Since aggregating all forest and non-forest samples to the national level negates the ability to assess the effect of TIC on classification accuracy under more specific physio-graphic or illumination conditions, we measured classification accuracies at three analytical scales: at the national level (i.e., using all available samples), by physiographic zone, and by IL stratum. For the physiographic zonation, we used three commonly used zones-the Mountains, Middle Hills, and Terai—that were delineated at the administrative district level by the Department of Survey Nepal in 1988 (http://rds.icimod.org/Home/DataDetail?metadataId=1597, accessed on 14 November 2020). Physiographic zones are topographically and societally relevant since they are used broadly for forest policy planning, implementation, and monitoring as well as national forest cover change mapping, and offer an analytical scale positioned in between the national level and the finely detailed IL stratum level. At each analytical scale, we examined differences in accuracies between nonTIC and TIC models with respect to the 95% confidence interval (CI) and offered explanations for any substantive similarities or differences.

*Objective 2: Measure the Effect of TIC on Long-Term Forest Cover Change*

We assessed the effect of TIC on long-term forest cover change by measuring differences in annual forest cover extent as well as the net amount and rate of forest cover change from 1992 to 2016 with nonTIC and TIC data across Nepal as well as by physiographic zone and IL stratum. To examine the potentially spatially and temporally varying differences between TIC and nonTIC results, we measured the differences (i.e., over or underestimate) between nonTIC and TIC forest cover extent and net forest cover change at every year and described how differences between TIC and nonTIC forest cover extent values vary over time at these three geographic levels. We also investigated whether TIC effectively normalized the distribution of net forest cover change across illumination conditions by measuring differences in the skew of nonTIC and TIC net forest cover change from 1992–2016 at the IL stratum level.

*Objective 3: Measure the Effect of TIC on Type of Forest Cover Change*

We assessed the effect of TIC on the type of forest cover change by measuring the area of land that was (a) Never Forest (i.e., stable non-forest from 1992–2016); (b) Always Forest (i.e., stable forest from 1992–2016); (c) Regenerated Forest (i.e., non-forest in 1992 but forest for at least one year between 1992–2016); and (d) Lost Forest (i.e., forest in 1992 but non-forest for at least one year between 1992–2016) using nonTIC and TIC data. We

measured differences in the respective amounts of these four kinds of conversions using nonTIC and TIC data at national, physiographic zone, and IL stratum levels.

## 3. Results

*Objective 1: Compare nonTIC and TIC Classification Accuracy*

Across various assessments of classifier model performance, TIC yielded a positive but small improvement (1.15%) on average classification accuracy at the national level (Table 2). The largest increase with TIC was found for the forest user's accuracy (3.25%), while TIC resulted in a decreased forest producer's accuracy of 0.69%. These results are generally in line with [30,39] that found modest gains in accuracy with TIC. Note, however, that the differences between nonTIC and TIC accuracies are usually within the respective 95% confidence intervals (CI) for nonTIC and TIC models. Across all measures, TIC has a lower CI, which suggests higher classification precision with TIC. The small difference in accuracies at the national level is likely a result of the training data being clustered in certain regions due to verification data availability, which may limit the variation of biophysical, physiographic, and illumination conditions being sampled. Furthermore, training data could only be sampled in regions that were sufficiently illuminated to visually discriminate forest from non-forest. This fundamental requirement, along with the fact that most of IL 10 lies above the tree line, effectively eliminated the potential to collect samples in the least-illuminated regions of the country where TIC would likely have the most improvement.

**Table 2.** Nepal-wide comparison of model accuracies (%) for nonTIC and TIC models with respective 95% CIs (in parentheses) based on three-fold cross-validation. OOB: Out-of-bag.

| TIC Model | Accuracy Measure | | | | | |
|---|---|---|---|---|---|---|
| | OOB | Validation | User's | | Producer's | |
| | | | Non-Forest | Forest | Non-Forest | Forest |
| nonTIC | 89.71 (0.38) | 87.05 (1.81) | 88.24 (4.83) | 85.44 (5.43) | 90.37 (5.17) | 82.17 (8.48) |
| TIC | 90.04 (0.21) | 88.39 (0.35) | 88.28 (3.02) | 88.69 (4.21) | 93.02 (3.23) | 81.48 (5.51) |

At the physiographic zone-level, the Mountains generally had the highest classification accuracies that were often 10% greater than corresponding measures in the Middle Hills (Table 3). The generally lower classification accuracies in the Middle Hills may be associated with this region's exceptional forest cover gain over the 25-year study period; with more conversions from non-forest to forest than other regions, the classifier seemingly had more difficulty capturing the resulting forested site. Even in the Mountains and the Middle Hills, zones that are characterized by a broad distribution of illumination conditions, differences between accuracies of nonTIC and TIC models remain small and rarely surpass either model's 95% CI. In the Mountains, TIC increases the producer's accuracy for forested samples by 2.38%, which is in excess of the CI. In the Middle Hills, TIC increases all accuracies except for the user's forest and producer's non-forest accuracies, even increasing the producer's accuracy for forest samples by 5.13%, but none of these accuracy changes exceed the respective CIs. Even in the highly illuminated Terai, the differences between nonTIC and TIC models across accuracy measures do not exceed the respective CIs.

Assessing accuracy differences by IL stratum reveals the same general pattern as above—that the differences in accuracies between nonTIC and TIC models rarely exceed the CIs—but spatial differentiation in TIC effects begins to be evident across the IL strata (Table 4). In less illuminated IL strata 2–6, the TIC model increased accuracies in 19 out of 30 measures where an additional 8 measures of accuracy are equal. For better illuminated IL strata 7–10, on the other hand, the relationship was inverted as TIC decreased accuracy in 18 of 24 measures. In contrast to the national and physiographic zone perspectives, OOB accuracy increases exceeded respective CIs for IL strata 4–6 and occasionally for other

accuracy measures across IL strata 3–6. The generally higher accuracies in less illuminated strata could be the result of more readily detected tree types in steeper slopes that typify low IL strata or, simply, a starker contrast between forest and non-forest land covers with increasing elevations and lower IL strata. It is unlikely that the higher accuracies are associated with the potential presence of snow cover since snow and ice were removed in pre-processing and forest cover classifications were based on growing season imagery. Nonetheless, assessing classification accuracy at the IL stratum-level accuracy is suggestive that TIC improves classification accuracy in low to moderately illuminated regions.

**Table 3.** Physiographic zone-level comparison of model accuracies (%) for nonTIC and TIC models with respective 95% CIs (in parentheses) based on three-fold cross-validation. OOB: Out-of-bag.

| Physiographic Zone | TIC Model | Accuracy Measure | | | | | |
|---|---|---|---|---|---|---|---|
| | | OOB | Validation | User's | | Producer's | |
| | | | | Non-Forest | Forest | Non-Forest | Forest |
| Mountains | nonTIC | 94.42 (0.68) | 92.48 (1.69) | 93.70 (0.79) | 90.35 (3.32) | 94.44 (1.81) | 89.12 (1.79) |
| | TIC | 94.91 (0.18) | 93.25 (1.63) | 94.72 (1.33) | 90.91 (2.33) | 94.33 (1.49) | 91.50 (2.06) |
| Middle Hills | nonTIC | 86.06 (0.39) | 82.79 (3.10) | 80.69 (4.98) | 85.45 (0.68) | 87.53 (0.80) | 77.75 (6.27) |
| | TIC | 85.97 (0.32) | 83.65 (3.41) | 83.29 (4.30) | 84.02 (4.29) | 84.41 (5.75) | 82.88 (4.74) |
| Terai | nonTIC | 92.01 (0.49) | 92.23 (0.26) | 92.33 (1.60) | 91.96 (5.44) | 96.85 (1.12) | 81.75 (1.94) |
| | TIC | 91.69 (0.55) | 91.59 (1.26) | 95.05 (2.91) | 84.21 (9.65) | 92.76 (4.98) | 88.89 (7.48) |

**Table 4.** Illumination condition stratum-level differences (i.e., TIC – nonTIC) in nonTIC and TIC model accuracies (%) with average 95% CIs (in parentheses) based on 3-fold cross-validation. Accuracy differences are presented for ease of comparison across ten strata. Note that IL stratum 1 is broadly above the tree line and was not sampled (i.e., N/A). OOB: Out-of-bag.

| IL Stratum | Differences in Accuracy Measure | | | | | |
|---|---|---|---|---|---|---|
| | OOB | Validation | User's | | Producer's | |
| | | | Non-Forest | Forest | Non-Forest | Forest |
| 1 | N/A | | | | | |
| 2 | 0.00 (0.00) | 0.28 (1.73) | 0.00 (0.00) | 0.28 (1.73) | 0.00 (0.00) | 0.00 (0.00) |
| 3 | 5.55 (0.00) | 7.16 (4.65) | 0.00 (0.00) | 7.55 (5.09) | 2.23 (1.25) | 0.00 (0.00) |
| 4 | 1.98 (0.03) | 1.24 (4.97) | 0.00 (0.00) | 1.58 (5.85) | 1.58 (2.90) | 0.00 (0.00) |
| 5 | 2.58 (0.00) | 0.63 (2.55) | 3.72 (7.21) | −0.67 (1.21) | −0.75 (3.40) | 1.51 (3.79) |
| 6 | 2.38 (1.13) | 3.35 (3.87) | 0.16 (7.83) | 5.17 (3.46) | 10.93 (4.26) | −1.27 (5.02) |
| 7 | −0.39 (0.69) | −1.50 (3.76) | 0.59 (3.98) | −3.77 (3.74) | −2.65 (2.24) | −0.23 (6.71) |
| 8 | −0.33 (0.46) | −0.23 (2.08) | 0.10 (2.62) | −1.15 (1.98) | −0.58 (0.90) | 0.52 (6.49) |
| 9 | −1.24 (0.60) | −0.92 (1.28) | 0.09 (1.36) | −3.80 (2.29) | −1.37 (0.77) | −0.35 (3.73) |
| 10 | 0.28 (0.76) | −0.66 (4.14) | −1.22 (4.80) | 0.05 (4.29) | −0.43 (3.81) | −0.73 (5.68) |

*Objective 2: Measure the Effect of TIC on Long-term Forest Cover Change*

Both nonTIC and TIC models show a steady expansion of forest cover across Nepal from 1992–2016 but differ in their estimates of annual net forest cover change (Figure 4). The nonTIC model (blue line) estimates a net forest cover gain of 29,960 km$^2$ (an average of

1248 km$^2$/year) while the TIC model (magenta line) estimates a lower net gain 28,076 km$^2$ (an average of 1170 km$^2$/year). The nonTIC model thus overestimates the TIC estimate of 1992–2016 net forest cover gain by 1884 km$^2$ (difference shown as black line), which corresponds to 1.3% of Nepal's land mass or 5.2% of Nepal's forest cover in 1992 as measured by the nonTIC model. Of special note, the direction of bias in the annual difference in forest cover area between nonTIC and TIC models inverts between 2000 and 2001. This flip from TIC over- to underestimating forest cover relative to nonTIC data is likely associated with the steady expansion and regeneration of forest cover into better-illuminated regions following well-documented agricultural abandonment and out-migration from the Middle Hills [49,53].

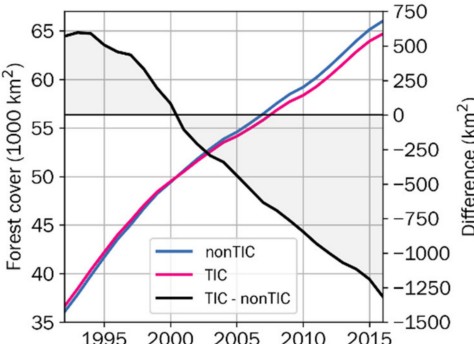

**Figure 4.** Annual forest cover area (left y-axis) based on nonTIC (blue line) and TIC (magenta line) models and difference (i.e., TIC – nonTIC; black line; right y-axis). Note the difference in scale between the two y-axes. While both nonTIC and TIC measures of forest cover area increase from 1992–2016, nonTIC exceeds the TIC estimate by over 500 km$^2$ at the beginning of the study. This difference diminishes until 2001, at which time the TIC estimate begins to exceed the nonTIC estimate, eventually capturing more than 1250 km$^2$ forest cover than the nonTIC estimate by 2016.

Moving to the physiographic zones, the Mountains, Middle Hills, and Terai each showed different spatial (Figure 5) and temporal (Figure 6) signatures revealing differences between nonTIC and TIC forest cover area measurements. At the beginning of the study period, the most pronounced difference in forest cover area was in the Terai, which saw 1.2% more forest cover with TIC. However, by the end of the study period, TIC underestimated Terai forest cover by 0.7% compared to the nonTIC model. The Middle Hills meanwhile had 0.4% more forest cover with TIC, but this bias inverted by 2016 where TIC found 1.4% less forest cover compared to the nonTIC model. By contrast, the TIC model in the Mountains consistently measured less forest cover than the nonTIC mode from 0.13% to 0.32% by the end of the study, though these differences are minor compared to the amount and overall flux of difference measured in the Middle Hills and Terai. At the national level, the Middle Hills and Terai expressed an inversion in bias 2001 and 2003, respectively, after which nonTIC exceeded TIC forest cover area. These findings echo the national assessment (Figure 4) and show that the effects of TIC on forest cover change measurements vary by physiographic conditions.

Differences in annual forest cover area and change between nonTIC and TIC models are much more pronounced at the IL stratum level than at physiographic (Figure 6) or national levels (Figure 4). TIC captured from 2% to 5.4% more net forest cover gain in less illuminated strata 1–6, and 1.3% to 3.3% less net forest cover gain in IL strata 7–10 (Figure 7). In effect, TIC rebalanced the distribution of net forest cover gain into both the less and moderately illuminated IL strata (i.e., IL 1–5) of Nepal. Looking at these IL strata differences over time, both TIC and nonTIC models show net forest cover gain each year from 1992–2016, but the better-illuminated strata 6–10 showed more rapid, early expansion in forest cover than less or moderately illuminated strata 1–5 (Figure 8a,b); this is likely the result of agricultural abandonment in the 1990s [49]. As TIC and nonTIC models captured regenerating forests differently in different strata, the nearly 13% range

in differences between forest cover area estimates in 1992 (i.e., IL 10 and IL 5) shrank to nearly 8% by 2016 (i.e., IL 1 and IL 4) (Figure 8c). TIC captured more net forest cover gain in IL 1–6 with the fastest gains occurring in the 1990s, which likely resulted from forest planting and expansion that started in the 1980s, while differences in IL 7–10 steadily decreased over the course of the study period as nonTIC captured ever more net forest cover gain in these strata. These results show the varying effects of TIC on forest cover measurement by year and by IL stratum, which are in part driven by specific kinds of forest cover change processes and patterns that vary by place and time. This is an important finding since it shows that different effects of TIC on forest cover change mapping are affected by illumination condition as well as the socioeconomic drivers at play.

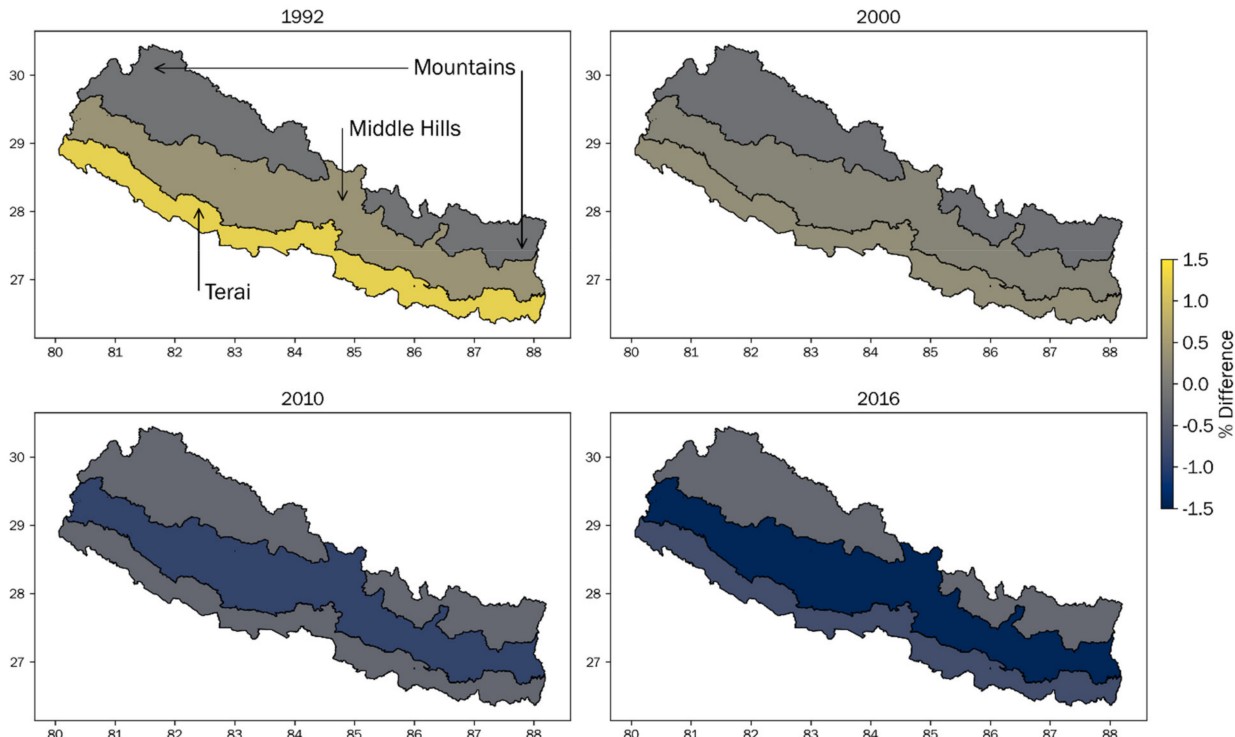

**Figure 5.** Variation in forest cover difference (i.e., TIC – nonTIC) expressed as percentage of area of Nepal's physiographic zones (Mountains, Middle Hills, and Terai) in 1992, 2000, 2010, and 2016. Across these four example years, the changing distribution of differences between nonTIC and TIC forest cover area estimates is apparent.

*Objective 3: Measure the Effect of TIC on Type of Forest Cover Change*

Compared to differences in model accuracy or the amount or rate of net forest cover change, larger differences between nonTIC and TIC are evident in measures of Never Forest (i.e., stable non-forest from 1992–2016); Always Forest (i.e., stable forest from 1992–2016); Regenerated Forest (i.e., non-forest in 1992 but forest for at least one year between 1992–2016); and Lost Forest (i.e., forest in 1992 but non-forest for at least one year between 1992–2016) (Figure 9). At the national-level, TIC estimates of Never Forest and Always Forest exceeded nonTIC estimates by 1.4% (1067 km$^2$) and 2.4% (744 km$^2$), respectively. The TIC model detected a more stable landscape with 4.6% (1637 km$^2$) less Regenerated Forest and 3.6% (174 km$^2$) less Lost Forest than the nonTIC model. Since the TIC model captured more forest cover in 1992 compared to the nonTIC model, it would be expected that the TIC model would see more Always Forest, which is indeed the case. However, the TIC model also measured more Never Forest. These results as well as the nonTIC model's detection of more net forest cover gain (Figure 4) speak to the rapid regeneration across the country over the 25-year period and show TIC yields a more conservative assessment of this forest cover change.

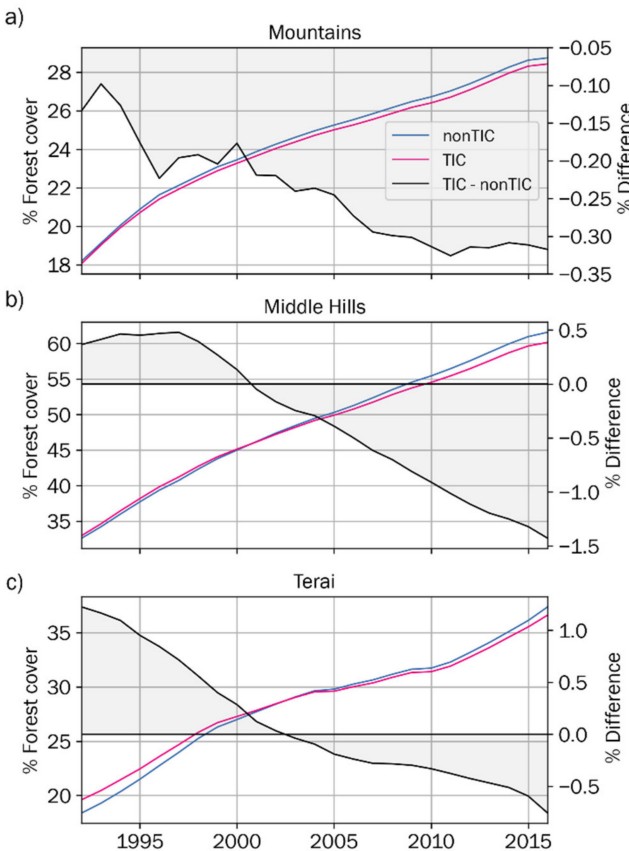

**Figure 6.** Annual forest cover area expressed as percentage of area of Nepal's physiographic zones (left y-axis) based on nonTIC (blue line) and TIC (magenta line) models and difference (i.e., TIC – nonTIC; black line; right y-axis). Note the difference in scale across all y-axes. The magnitude and rate of change in the difference between nonTIC and TIC estimates of forest cover change differently vary between the three physiographic zones due to differing rates of forest cover regeneration.

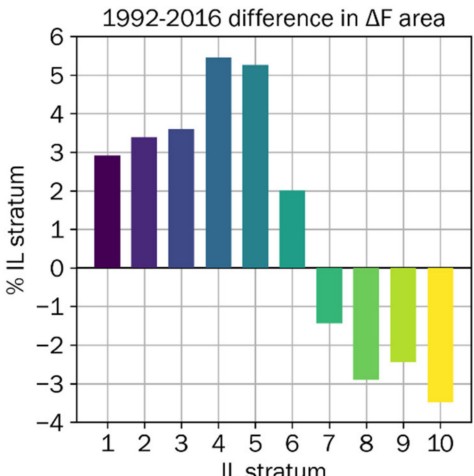

**Figure 7.** Distribution of the difference (TIC – nonTIC) in net forest cover from 1992–2016 by IL stratum. The TIC approach detects more forest cover gain from 1992–2016 in IL strata 1–6, while the nonTIC approach detects more forest cover gain in IL strata 7–10.

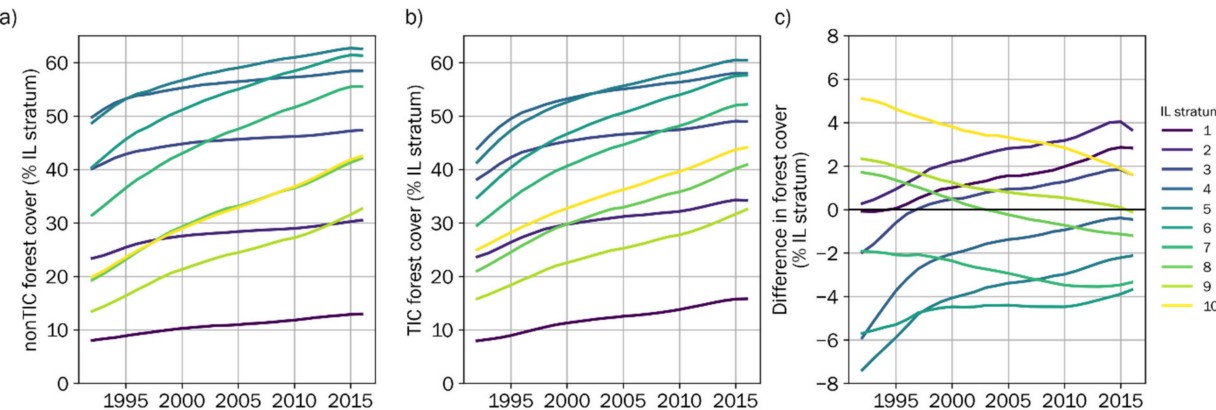

**Figure 8.** Annual forest cover area estimates for (**a**) nonTIC and (**b**) TIC models and (**c**) differences (i.e., TIC – nonTIC) by IL stratum from 1992–2016. IL strata show divergent trends of difference in forest cover area estimates over time, a result of differentiated rates of forest cover gain within each stratum.

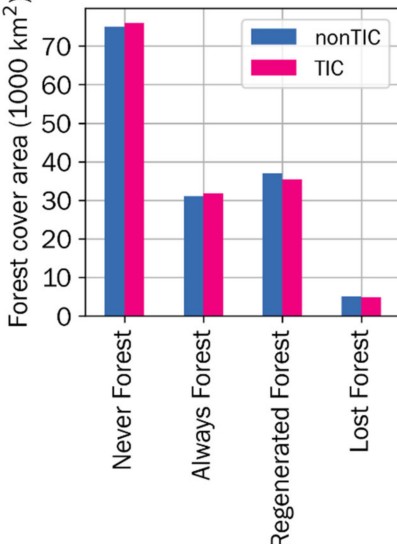

**Figure 9.** Comparison of national-level forest cover conversion areas between nonTIC and TIC models. The nonTIC model shows more total regeneration and loss than the TIC model between 1992–2016.

At the level of physiographic zones, the Mountains and Terai are primarily characterized as Never Forest by both models (approximately 68% and 57%, respectively) while the Middle Hills is rather balanced between Regenerated Forest (approximately 34%), Never Forest (33%), and Always Forest (29%) with very little Lost Forest (4%) (Figure 10a). The Middle Hills is thus a much more dynamic region with regard to long-term forest cover gain compared to the Mountains and the Terai. However, the largest differences between the nonTIC and TIC models in their estimation of forest cover conversion are found in the Terai, which has a 3.1% range of difference defined by TIC's 1.2% overestimate of Always Forest and −1.9% underestimate of Regenerated Forest compared to nonTIC (Figure 10b). The Middle Hills, meanwhile, had a 2.4% range of difference between Never Forest and Regenerated Forest, while the Mountains showed very little difference in the four conversion estimates with a total range of 0.5% across the four conversions. TIC thus generally overestimates Never Forest and Always Forest, and underestimates Regenerated Forest and Lost Forest compared to nonTIC, but TIC and nonTIC never disagree by more than 1.9% (i.e., in the case of Terai Regenerated Forest).

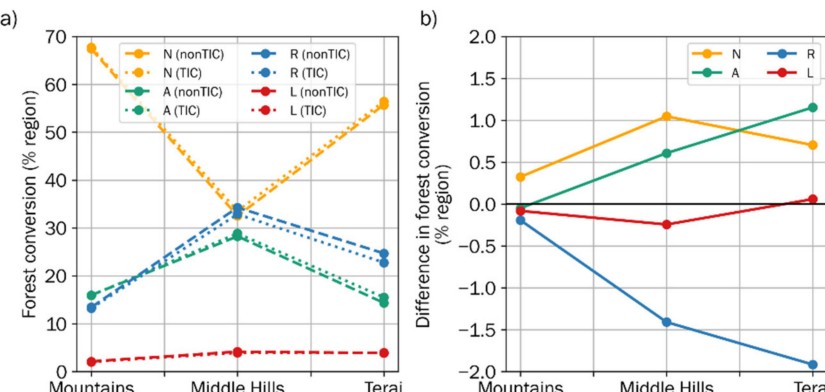

**Figure 10.** (**a**) Area and (**b**) difference (i.e., TIC – nonTIC) of Never Forest (N), Always Forest (A), Regenerated Forest (R), and Lost Forest (L) of nonTIC and TIC models by physiographic zone. The differences in forest cover conversions across physiographic zones are associated with varying drivers and rates of forest cover change.

Both nonTIC and TIC models measured Regenerated Forest at 10–30% in each IL stratum with the most Regenerated Forest in IL 6–8 (Figure 11a); the sun-oriented and not-too-steep topography of these IL strata suggests that forest regeneration followed agricultural abandonment here. The models nonetheless showed considerable disagreement with TIC measuring 4–6% more Regenerated Forest in IL 1–5 and 2–3% less Regenerated Forest in IL 8–10 than the nonTIC model (Figure 11b). Lost Forest, by comparison, was consistently low across IL strata with a mean of 3.5% per stratum for nonTIC and TIC alike, and very little difference (<1%) between models across IL strata. Considering stable conversion classes, Never Forest pervaded the lowest (1–3) and highest (7–10) IL strata, while Always Forest dominated IL strata 4–5 (Figure 11a). The TIC model detected 4–5.2% less Never Forest in IL 1–2 but >3% more Never Forest in IL 6–7 than the nonTIC model. The TIC model consistently detected less Always Forest than the nonTIC model from IL strata 1–7, since these strata were broadly characterized as Regenerating Forest or net forest cover gain by the TIC model. Thus, the effect of nonTIC on forest cover conversion not only varies by IL strata; it also varies by conversion type: nonTIC sees more Regenerated Forest in IL strata 1–5, more Never Forest from IL strata 6–7, and more Always Forest from IL strata 8–10. In contrast to national-level forest conversion trends that show TIC capturing more forest cover stability (principally due to the very large areas of IL strata 8–10, Figure 2), these IL stratum-level views of forest conversion show that TIC has a highly variable effect for quantifying both forest cover stability and change.

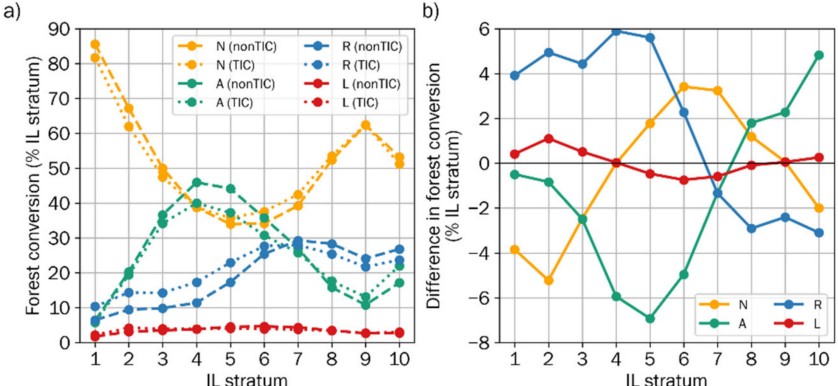

**Figure 11.** (**a**) Area and (**b**) difference (i.e., TIC – nonTIC) of Never Forest (N), Always Forest (A), Regenerated Forest (R), and Lost Forest (L) of nonTIC and TIC models by IL stratum. The differences in forest cover conversions across IL strata are associated with varying drivers and rates of forest cover change.

## 4. Discussion

This study offers several novel insights towards understanding the effects of TIC on measurements of forest cover area and change. First, our cross-scale assessment of classifier accuracy showed that the TIC model yielded only modest accuracy gains at the national level, which were broadly negligible at the physiographic zone and IL stratum levels. Second, our use of a 25-year-long time series allowed us to show that TIC does not have a stable effect on forest cover mapping but rather varies over time. TIC resulted in a more conservative assessment of net forest cover gain and forest cover conversion over the entire study period of 1992–2016. However, these trends are not consistent over the full 25-year-long study period. From 1992–2001, TIC captured more annual forest cover than the nonTIC model at the national level (Figure 4) but, from 2001–2016, nonTIC captured more forest cover than TIC. Third, TIC effects on measures of forest cover and change show considerable variation by physiographic zone and IL stratum. For example, the larger increase in forest cover gain from 1992–2001 measured with TIC was partially driven by exceptional forest cover expansion in IL strata 4–5 (Figure 12a). By 2001–2016, forest cover expansion in IL strata 4–5 and the Middle Hills had waned, resulting in more similar change estimates between TIC and nonTIC (Figure 12b). Taken together, these findings show that even if classification accuracies between nonTIC and TIC are comparable, nonTIC approaches may alternately under- or overestimate forest cover area and net forest cover change depending on the period under investigation as well as the regional physiographic and illumination conditions. The temporally divergent effects of TIC on forest cover area measurements were only detectable through using a multi-decadal annual time series and are especially important to consider for accurate documentation of long-term forest cover change in mountainous or otherwise topographically complex regions.

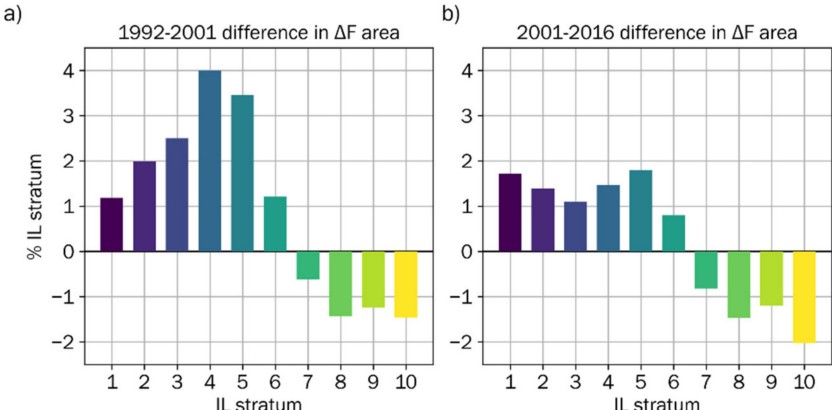

**Figure 12.** Difference (i.e., TIC – nonTIC) in forest cover change from (**a**) 1992–2001 and (**b**) 2001–2016. Differences in forest cover change estimates between nonTIC and TIC models result from differences in model sensitivity to forest cover under different illumination conditions as well as the rate of forest cover change in different time periods.

This study also shows that TIC not only consistently detects additional forest cover (up to 4%) in the least-illuminated strata (IL 1–2), but also that TIC detects more forest cover (up to 5.1%) in the most-illuminated strata (IL 9–10) for all but the last 2 years of the study. Indeed, as the study progressed from 1992–2016, TIC measured an increasingly larger area of forest cover than the nonTIC model in low IL strata and increasingly less forest cover in high IL strata. These annual forest cover differences translated into TIC detecting more net forest cover gain (3–5.5%) and Regenerated Forest (4–6%) in low and moderately illuminated strata (IL 1–5) than the nonTIC model and less forest cover gain in the most-illuminated strata (IL 7–10). The additional forest cover gain in low IL strata captured in the TIC model helped to balance the net forest cover change across IL conditions. The ultimate effect of TIC helping to normalize the distribution of forest cover gain highlights the importance of examining spatially divergent effects of TIC on forest cover change mapping

at different stages of a forest transition. In the absence of using TIC in mountainous or otherwise topographically complex regions, scholars should give pause to net forest cover change results that are positively skewed towards better illuminated regions.

The various ways, summarized above, in which TIC can affect forest cover area and change mapping are directly related to shifts in forest cover change drivers over time. In Nepal, two of the most important indirect drivers of forest cover change are community forest management and migration [53,69]. In 1988, 61% of Nepal's forested area (estimated at 3.5 million ha) was designated by the Department of Forests (DoF) as being eligible for community management, which decentralized access, usage, and management rights of the forest to community user groups [70]. In the intervening decades, many scholars have identified the positive effect of Nepal's community forest management on the conservation and expansion of forest cover within the country through local interview data collection, participatory mapping, and regional satellite image analysis (e.g., [49,53,69]). The expansion of community forestry, especially in the Middle Hills in the 1980s, brought reduced cutting and increased tree planting, which drove the rapid expansion of forest cover in the 1990s detected by the TIC model (Figure 12a). By the 1990s, the selective curtailing of timber harvesting led to forest cover starting to expand in regions that had not been forested for decades [71]. In the early 2000s, the Maoist insurgency and political disorder following the 2001 assassination of the king drove widespread out-migration from the Middle Hills and Mountains, which led to forest cover gain following the abandonment of rainfed agricultural lands and the great decline of tree harvesting in these regions [46]. Given the history of shifting importance of forest cover change drivers, research that seeks to understand the influence of drivers on the spatial pattern and timing of forest cover change would also do well to use TIC on long-term forest cover change measurements.

The findings of this study thus reinforce the need for adopting TIC for policy-relevant mountainous forest cover mapping since nonTIC results may suggest a more rapid or otherwise effective intervention. For example, the nonTIC model measured 1884 km$^2$ more net forest cover gain across Nepal from 1992–2016 compared to the TIC model (Figure 4). This overestimation of net forest cover gain at the national level is important to recognize when assessing the outcomes of national forest cover conservation efforts or REDD+ initiatives in Nepal [72,73]. Similarly, TIC should be applied when collecting empirical data on forest cover or change to examine pathways of forest recovery such as resource scarcity or economic development [74–77]. Evidence of resource scarcity and economic development vary by spatial scale, location, and time, all of which benefit from annual, long-term assessments of forest cover change and conversion dynamics [78]. However, to date, TIC is rarely used in satellite-based studies of forest transition in Nepal (e.g., [48]) and other mountainous forested regions.

Future research examining the effect of TIC on satellite image-based forest cover and forest cover change mapping would do well to examine in more detail the changes that TIC introduces to time-series segmentation and trend fitting that were not considered in this study. While time-series segmentation and trend fitting help to build a spectrally stabilized profile for training data generation and classification, they may minimize some of the change in spectral condition caused by TIC. This study also used a long-term mean IL across all input imagery to stratify the landscape and summarize results, but did not examine areas with high inter-annual variation in IL, which may affect both TIC and nonTIC measurements of forest cover presence. Future work could look at the role of IL variation on forest cover classification accuracy and derived measures of change in more detail, and could examine whether TIC effects on Random Forest-classified forest cover are consistent when using other classification approaches. An examination of TIC effects using object-based classification approaches would also be warranted since TIC may affect image segmentation in ways that are not relevant for consideration using the pixel-based classifier used here. Finally, the late morning overpass time of Landsat, Sentinel-2, and commercial Earth observing satellites (including those commonly used to generate high resolution reference imagery) means that we commonly lack imagery with illuminated

west-facing slopes. Future research on forest cover mapping in topographically complex environments would thus benefit from incorporating imagery acquired at different times showing more diverse illumination conditions.

## 5. Conclusions

This study characterized the previously undocumented effects of TIC on forest cover and forest cover change mapping in Nepal from 1992–2016 using Landsat surface re-flectance time series. We developed parallel TIC and nonTIC assessments on classifier model accuracy, long-term trends, and forest cover conversion and quantified the differences introduced by TIC. We summarized results at national, physiographic zone, and IL stratum levels and over various time periods to account for the ways in which TIC affects our understanding of forest cover change in different regions at different times due to different socioeconomic drivers. Compared to the nonTIC approach, we found that TIC modestly improves classifier accuracy by an average of 1.15% and measured 1884 km$^2$ less net forest cover gain than the nonTIC approach, which is equivalent to 1.3% of Nepal's land area. However, the TIC approach resulted in more forest cover gain in less-illuminated regions (IL strata 1–5), effectively helping to normalize measures of forest cover change regardless of illumination conditions. Mountainous forest cover change mapping without using TIC thus risks overestimating forest cover gain in well illuminated strata (by 1–3% in Nepal) and underestimating forest cover gain in darker regions (by 3–5% in Nepal). In countries such as Nepal where different drivers (e.g., forest policy, out-migration, and social conflict) impact forest cover change differently over space and time, the spatially and temporally harmonized estimates provided by TIC offer more quantitative accuracy in forest cover change measurements at the national and local scale, and support a more confident interpretation of the reasons for and consequences of forest cover change.

**Author Contributions:** Conceptualization, J.V.D.H.; methodology, J.V.D.H., A.C.S. and K.H.; software, J.V.D.H., A.C.S. and K.H.; validation, J.V.D.H. and A.C.S.; formal analysis, J.V.D.H., A.C.S. and K.H.; writing—original draft preparation, J.V.D.H., A.C.S., K.H., S.S. and J.F.; writing—review and editing, J.V.D.H., A.C.S., K.H., S.S. and J.F.; visualization, J.V.D.H.; funding acquisition, J.F. All authors have read and agreed to the published version of the manuscript.

**Funding:** This study was funded by the National Aeronautics and Space Administration's (NASA) Land-Cover and Land-Use Change Program (LCLUC) Grant No. NNX15AF65G. We thank NASA's LCLUC Program for its support.

**Institutional Review Board Statement:** Not applicable.

**Informed Consent Statement:** Not applicable.

**Data Availability Statement:** Not applicable.

**Acknowledgments:** We thank Robert Kennedy, Zhiqiang Yang, and the Google Earth Engine user community for their assistance in developing this study. We also acknowledge the Ampinefu ("Mary's River") band of the Kalapuya people who are the original inhabitants of the land now occupied by Oregon State University.

**Conflicts of Interest:** The authors declare no conflict of interest.

## Appendix A

**Table A1.** Annual and monthly distributions of 1893 Landsat 5, 7, and 8 images used in study, collected from July–October from 1992–2016.

| Year/Month | July (7) | August (8) | September (9) | October (10) |
| :---: | :---: | :---: | :---: | :---: |
| 1992 | 4 | 8 | 11 | 12 |
| 1993 | 10 | 13 | 7 | 14 |
| 1994 | 13 | 9 | 17 | 23 |

**Table A1.** *Cont.*

| Year/Month | July (7) | August (8) | September (9) | October (10) |
|:---:|:---:|:---:|:---:|:---:|
| 1995 | 3 | 7 | 2 | 6 |
| 1996 | 9 | 7 | 20 | 20 |
| 1997 | 6 | 11 | 12 | 16 |
| 1998 | 4 | 7 | 17 | 19 |
| 1999 | 7 | 7 | 14 | 21 |
| 2000 | 9 | 11 | 23 | 24 |
| 2001 | 16 | 20 | 18 | 12 |
| 2002 | 7 | 9 | 11 | 18 |
| 2003 | 6 | 7 | 3 | 14 |
| 2004 | 14 | 13 | 26 | 36 |
| 2005 | 7 | 13 | 25 | 30 |
| 2006 | 8 | 11 | 35 | 32 |
| 2007 | 7 | 12 | 19 | 23 |
| 2008 | 11 | 10 | 27 | 47 |
| 2009 | 21 | 15 | 35 | 44 |
| 2010 | 8 | 10 | 17 | 41 |
| 2011 | 14 | 20 | 23 | 37 |
| 2012 | 2 | 10 | 19 | 29 |
| 2013 | 20 | 26 | 48 | 46 |
| 2014 | 20 | 30 | 38 | 53 |
| 2015 | 32 | 25 | 50 | 50 |
| 2016 | 13 | 39 | 37 | 51 |
| **Total** | 271 | 350 | 554 | 718 |

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
