# Peer review of "Shedding New Light on Mountainous Forest Growth: A Cross-Scale Evaluation of the Effects of Topographic Illumination Correction on 25 Years of Forest Cover Change across Nepal"

_remotesensing, doi:10.3390/rs13112131_

Round 1
Reviewer 1 Report
General comments:
The authors of this study compare the use of topographic illumination correction algorithms vs non-use in forest cover classification, forest cover change and forest cover change geographic distribution using the country of Nepal as the study area and Landsat imagery time series imagery.
The study is overall well written with an attention to detail especially in the material and methods and results section, which makes the study very clear. Also as the study mentions, the topography of the study area(Nepal) represents a good choice of the TIC potential. The abstract is good and comprehensive.
Some small improvements can be done. In the discussion you mention only comparisons with non-TIC, but no mention of the improvements/differences in terms of accuracy. Maybe a few sentences are warranted. Also, the fact that you used Random Forest classifier with the mentioned specifications can have a significant impact on end results. You mention that at then of discussion the need for further work, but maybe a little bit more can be added.
Specific comments:
L61-61: strange phrasing
Modify the text in the discussion from figure 13 to figure 12(figure 13 does not exist).
L521: reference to figure 13a should be 6a?
Author Response
Some small improvements can be done. In the discussion you mention only comparisons with non-TIC, but no mention of the improvements/differences in terms of accuracy. Maybe a few sentences are warranted. Also, the fact that you used Random Forest classifier with the mentioned specifications can have a significant impact on end results. You mention at the end of the discussion the need for further work, but maybe a little bit more can be added.
--> Thank you for your review, and for the suggestion to clarify the changes in accuracy with TIC as well as the Random Forest parameters in the Discussion. We have made several improvements to the first (starting on line 483) and last paragraphs (starting on line 561) of the Discussion accordingly.
Specific comments:
L61-61: strange phrasing
--> We have revised this sentence to hopefully make it clearer.
Modify the text in the discussion from figure 13 to figure 12(figure 13 does not exist).
--> Thank you for catching this. We have corrected the figure numbering.
L521: reference to figure 13a should be 6a?
--> This connects to Fig. 12a. We have corrected.
Reviewer 2 Report
It is very strange to analyze forest cover in areas above 4000 m, which is above the tree line where forest cover ceases to exist. The work could have been minimized by taking a considerable number of samples by masking the area above the tree line using elevation data. Strangely, the middle hills include snow-covered areas which are above 4000 m asl (Figure 1a). It is also very strange to have considerable illumination variations in the Terai plain, which is known as the flat areas lying south of the Himalayan foothills (Siwalik hills). I suggest that you improve your physiographic map. In a mountainous country like Nepal, illumination variation is unavoidable, which affects classification accuracy. Thus, your research will be more valuable if you stress the usefulness of topographic illumination correction for assessing forest cover than giving an elaborate explanation on illumination stratum-level differences.
Detail comments:
- What is the motivation of your research? What do you want to find out? It will be better to list some research questions.
- The specific objectives are clear but what is the main objective of the research?
- Lines 98-101
On what basis you have separated mountains from the hills. Provide some details on the physiographic zonations of Nepal. I suggest that you take a look at the work of the Land Resources Mapping project of Nepal, undertaken by the Government of Nepal and Canada (https://issuu.com/manohardhami00/docs/land_resource_mapping_report_summar).
- Figure 1
Strangely, the middle hills include snow-covered areas which are above 4000 m asl. Please check.
- Lines 135
It will be useful to provide a list of images with acquisition month and year
- Equation 1
Provide a reference for this equation
- Figure 2e
Generally speaking, in the plain areas illumination should be normally uniform. It is very strange to have still quite some variations in topographic illumination in the Terai plains. Explain your reason.
- Lines 187 – 198
If the images you have used for analysis were cloud-free (see lines 135), I do not understand why you need to calculate the proximity of the pixels to clouds or use NIR reflectance to avoid shaded or clouded pixels.
- Lines 211 – 214
- What are the added value of using all the products like NDVI, EVI, and Greenness, Tasseled cap, etc. when they are generated using the same spectral bands.
- Lines 214-218
Your mountainous areas are the ones above 4000 m asl, which is above the tree lines. There should be no forest cover above the tree lines. What is your explanation for this?
- Lines 222-224
- A pixel can be either forest or non-forest. How it is possible to be a 50% forest canopy? It is not clear.
- Lines 225 – 231 and Figure 3a
You could have separated non-forest canopy in high altitude areas using the concept of tree lines and SRTM data. This would have reduced your visual interpretation efforts tremendously.
- Lines 231-233 and Figure 3b
I am surprised to see forest cover in your physiographic unit Mountains, which falls above the tree line.
- Lines 239-240
Explain what is one-thousand trees in out-of-bag mode.
- Lines 302 -305
This is possible if you have used the same test samples to check classification accuracy for all the periods. Clarify this.
- Lines 322-324
Is this because of snow cover?
- Lines 322-323 and Table4
Explain what could be the reasons for the decrease of classification accuracy in high illuminated areas? Is it more so in the mountains and in the middle hills?
- Lines 326-329
Steeper slopes do not necessarily mean low illumination. It depends on the direction of the slope (aspect) with respect to the position of the sun (sun elevation). In Nepal, the north-facing slopes have generally low illumination (or even shadow when the elevation of the ridge is high) and the south-facing slopes have relatively high illumination while both the north-facing and the south-facing slopes could steep. It is not clear how you have used the slope direction in your analysis.
- Figure 5
Instead of showing variations in forest cover differences (e.g. TIC and nonTIC), it will be more useful for the readers to show forest cover situations in different periods.
- Figure 8
I do not understand the usefulness of this figure. Of course, the results show divergence of forest cover estimates in each period. What can you conclude from this and what is the use of this result?
- Lines 423-425 and Figure 9
This figure is interesting. I find it very strange to see that regenerated forest cover is larger than always forest cover. How do you define regenerated forest? Can regenerated forest occur inside the always forest area?
Author Response
We thank Reviewer 2 for their general and detailed comments. Please see the PDF attachment with our replies to the reviewew's report with figures included.

Reviewer 3 Report
The MS investigates the effects of TIC on forest cover mapping in Nepal from 1992 – 2016 using Landsat imageries. An important study to understand the impact of TIC on measurements of forest cover area and change in Nepal. This study's finding is essential to the growing number of forest cover-related studies from Nepal and worldwide. MS is well prepared. Few minor suggestions are below:
Line 133. Please adjust the reference according to the journal format
Line 233. I suggest using forest line instead of tree line here.
Line 479, 481. Fig 13a or Fig. 12a?; Fig. 13b or Fig. 12b? Please correct this for the entire MS
Line 533. km2
Author Response
Thank you for your review.
Line 133. Please adjust the reference according to the journal format
--> Thank you for catching this. We have fixed the reference format.
Line 233. I suggest using forest line instead of tree line here.
--> Thanks; corrected.
Line 479, 481. Fig 13a or Fig. 12a?; Fig. 13b or Fig. 12b? Please correct this for the entire MS
--> Thanks for catching this. Fixed.
Line 533. Km2
--> Thanks for catching this. Fixed.